# Metabolic Myopathies in the Era of Next-Generation Sequencing

**DOI:** 10.3390/genes14050954

**Published:** 2023-04-22

**Authors:** Jon Andoni Urtizberea, Gianmarco Severa, Edoardo Malfatti

**Affiliations:** 1Institut de Myologie, la Pitié-Salpétrière Hospital, 75013 Paris, France; 2Department of Medical, Surgical and Neurological Sciences, Neurology-Neurophysiology Unit, University of Siena, Policlinico Le Scotte, Viale Bracci 1, 5310 Siena, Italy; 3Université Paris Est, U955, IMRB, INSERM, APHP, Centre de Référence de Pathologie Neuromusculaire Nord-Est-Ile-de-France, Henri Mondor Hospital, 94000 Créteil, France

**Keywords:** muscle metabolism, metabolic myopathies, glycogen storage disorders, muscle glycogenoses, lipid storage diseases, Pompe disease, McArdle disease, Cori–Forbes disease, Tarui disease, polyglucosan body myopathies, TK2 myopathy, mitochondrial myopathies

## Abstract

Metabolic myopathies are rare inherited disorders that deserve more attention from neurologists and pediatricians. Pompe disease and McArdle disease represent some of the most common diseases in clinical practice; however, other less common diseases are now better-known. In general the pathophysiology of metabolic myopathies needs to be better understood. Thanks to the advent of next-generation sequencing (NGS), genetic testing has replaced more invasive investigations and sophisticated enzymatic assays to reach a final diagnosis in many cases. The current diagnostic algorithms for metabolic myopathies have integrated this paradigm shift and restrict invasive investigations for complicated cases. Moreover, NGS contributes to the discovery of novel genes and proteins, providing new insights into muscle metabolism and pathophysiology. More importantly, a growing number of these conditions are amenable to therapeutic approaches such as diets of different kinds, exercise training protocols, and enzyme replacement therapy or gene therapy. Prevention and management—notably of rhabdomyolysis—are key to avoiding serious and potentially life-threatening complications and improving patients’ quality of life. Although not devoid of limitations, the newborn screening programs that are currently mushrooming across the globe show that early intervention in metabolic myopathies is a key factor for better therapeutic efficacy and long-term prognosis. As a whole NGS has largely increased the diagnostic yield of metabolic myopathies, but more invasive but classical investigations are still critical when the genetic diagnosis is unclear or when it comes to optimizing the follow-up and care of these muscular disorders.

## 1. Introduction

Classically, metabolic myopathies encompass glycogen storage disorders (GSD), fatty acid oxidation (FAO) defects and mitochondrial disorders. All have a varying degree of muscle involvement due to a faulty protein involved in the intermediary metabolism of glucose and fatty acids [1].

Rare and even ultra-rare metabolic myopathies represent a clinically and genetically heterogeneous group of muscular conditions. From a clinical perspective, one should distinguish between the metabolic myopathies with static or progressive muscle weakness, such as GSD II, GSD III, and GSD IV, and those for which the symptomatology is acute and relapsing (dynamic), with exercise-induced myalgia with or without rhabdomyolysis, such GSD V and other GSDs, FAO defects, and mitochondrial myopathies [2]. However, realistically, the boundary between these two groups is sometimes challenging to delineate.

Genetic testing has become the first line of investigation for genetic myopathies over the years, and the field of metabolic myopathies is no exception. Historically, muscle histology and biochemical assays of all kinds and in various tissues have been the pillars used to phenotype and categorize metabolic myopathies. The advent of next-generation sequencing (NGS) led to a paradigm shift, enabling the simultaneous screening of a great number of genes [1,3], also representing an opportunity, from a research standpoint, to unravel novel disease-causing genes in some unsolved cases. The flip side of the coin is that NGS often provides a great number of variants, such as VUS (variants of uncertain significance) that are difficult to interpret and to validate. Data analysis is particularly challenging when the clinician is faced with unspecific complaints such as isolated chronic myalgia or certain laboratory abnormalities such chronic serum CK elevation below 1000 IU/L.

In these scenarios, the classic diagnostic algorithm including myopathology and in-tissue enzyme assays remains fundamental to confirm the diagnosis, especially in treatable conditions.

Another source of excitement in metabolic myopathies is the therapeutic advances that a growing number of patients have been benefitting from [1]. Beyond interventions at the symptomatic level, an increasing number of molecules can target the primary cause of the disorder. Promising advances have also been achieved recently in the clinical development of next-generation enzyme replacement therapy (ERT) and gene replacement approaches [4,5]. Based on the principle that early interventions are preferable to secure better efficacy, several metabolic myopathies are now routinely screened neonatally in many places [6].

## 2. General Considerations

### 2.1. Overview of Muscle Metabolism

Muscles utilize energy in order to produce muscle contractions. The main consumables used to deliver energy within muscle fiber derive from the adenosine triphosphate (ATP) and creatine phosphate systems. The cleavage of immediately available glucose molecules (glycolysis) is key at the initiation of exercise. The muscle then switches to a more sustainable source of energy such as carbohydrates and, later, lipids. The breakdown of glycogen (glycogenolysis), a carbohydrate polymer, is essential to provide glucose to the muscle fiber, which is activated within two seconds of muscle contraction. The flux through glycolysis is generated by glucose-6-phosphate (G-6-P), which is provided by glycogenolysis fatty acids, the primary energy source for muscle at rest and during periods of prolonged low-intensity exercise. Short- and medium-chain fatty acids enter the mitochondrion freely. The longer chains (LCFA and VLCFA) require a specific mode of transportation through the inner mitochondrial membrane. This transfer is controlled by at least three proteins: acylcarnitine translocase, carnitine palmitoyltransferase (CPT) I, and CPT II. Fatty acids then undergo a process of catabolization called β-oxidation within the mitochondrion, coupled with the respiratory chain and its oxidative phosphorylation (OXPHOS) machinery. The ratio of fat to carbohydrate oxidation is a function of anaerobic threshold, training status, sex, and percentage of VO2 max. Figure 1 provides an overview of the altered pathways in metabolic myopathies. For a more comprehensive review of the muscle energetic metabolism, refer to [7].

### 2.2. Epidemiology

Even if grouped, all metabolic myopathies remain a class of rare diseases, many of which are ultra-rare. Most of them are autosomal recessive, but maternally inherited myopathies (of mitochondrial origin) and even rarer autosomal dominant and X-linked metabolic myopathies do exist. Recessive myopathies are more prevalent in endogamous areas (Maghreb, the Middle East, and the Indian subcontinent) or in isolates (French Guiana) [8,9]. However, the presence of GSD II seems to be less prevalent in the Middle East than originally expected [8]. This might be due to an ascertainment bias and/or a poor recognition of the disease. Founder mutations or predominant variants have been reported notably in GSD II, as well as in McArdle disease [10]. Interestingly, numerous cases of riboflavin-responsive MADD (RR-MADD) have been reported in Far East Asia and in Iran [11]. The prevalence of TK2 deficiency seems to be higher in Spain and in Hispano-Americans, suggesting the existence of a founder effect [12].

Life is usually compatible with most metabolic myopathies, with notable exceptions, such as early-onset FAO defects; lipin-1 deficiency; and life-threatening metabolic crises, including lactic acidosis, severe hypoglycemia, and rhabdomyolysis [2]. Cardiac arrest and sudden death also need to be considered and therefore prevented in adults diagnosed with specific metabolic disorders.

Some authors have noted that statins, vitamin D deficiency, and hypothyroidism may lower the threshold of clinical manifestations in metabolic myopathies [1]. Hence, there is a need to detect, document, and prevent these external factors or comorbidities in every patient presenting with a metabolic myopathy.

### 2.3. NGS in Metabolic Myopathies: Advantages and Challenges

NGS diagnostic testing includes gene panels, often with the most widely used first, followed by whole-exome sequencing (WES) and whole-genome sequencing (WGS) [13]. A total of 30 nuclear genes have been associated with metabolic myopathies to date [14]. M Kerr et al. showed that mitochondrial disorders that can be related to either mitochondrial or nuclear genome variants should be initially investigated using a non-invasive, bigenomic sequencing (BGS) approach (using both an nWES and optimized mtDNA analysis to include large deletions). The usage of tissue biopsies remains prevalent in the event of unsolved cases or when the diagnosis is still not clear after NGS studies [15].

For variant analysis, the application of the American College of Medical Genetics and Genomics (ACMG) criteria elaborated in 2015 is required [16]. Interpretation of VUS is particularly challenging and inconsistent, as with the progress of knowledge can be both upclassified or downclassified. Nevertheless, when considering all genetic conditions, over time, there is an approximate 75% chance that a VUS is downgraded and an approximate 25% chance that a VUS is upgraded [17].

The detection of a VUS requires additional tests to verify its clinical relevance: first, a familial segregation study, followed by a detailed clinical reassessment to better characterize the phenotype, a novel electrophysiological study, and, finally, myopathological studies with immunostaining and Western blotting (WB). These tests can further functionally validate the variants found in novel genes [13].

In the event of incidental findings during genetic studies, the management of communication is followed closely in relation to the patient’s preference expressed in the informed consent, which contains a specific section with respect to incidental findings according to specific national legislation.

The use of NGS panels has some disadvantages, as this kind of technology is not able to analyze certain parts of the human genome, for example, repeated sequences, highly homologous regions, and regions of high and low GC content. Therefore, current NGS panels are not able to detect various neuromuscular disorders, such as myotonic dystrophy type 1 and 2, oculopharyngeal muscular dystrophy (OPMD), oculopharyngodistal myopathies, and facioscapulohumeral muscular dystrophy (FSHD1) [13]. Nevertheless, no report on metabolic myopathies linked with expansion of repeated elements has been published to date.

In recent years, disease-modifying polymorphisms have been described in few metabolic myopathies. For example, in 2020, an Italian multicenter study showed that *ACE* and *ACTN3* polymorphic genes can influence diaphragmatic dysfunction in patients with LOPD [18].

### 2.4. Clinical Presentation

Figure 2 depicts clinical symptoms and/or signs highly suggestive of metabolic myopathies, such as bouts of rhabdomyolysis, myalgia, cramping, exercise intolerance, and myoglobinuria/pigmenturia. Some patients present fewer specific symptoms and/or signs, such as fixed or progressive weakness affecting pelvic, scapular, axial, and/or limb musculature [19]. When present, muscle weakness may vary from mild to generalized, resulting in life-threatening hypotonia and respiratory insufficiency.

Apart from these suggestive presentations, the clinician may occasionally be misled by extramuscular associated signs in isolation or added to the general picture, such as cardiomyopathy, respiratory compromise, ptosis, ichthyosis, hepatopathy, and encephalopathy [19].

Each symptom, notably muscle pain and exercise-induced weakness, must be dissected and documented carefully. The kinetics of exercise intolerance must be investigated. The concepts of second-wind and out-of-wind phenomena are key in myophosphorylase deficiency (McArdle disease) and phosphofructokinase deficiency (Tarui disease) [20], respectively. The circumstances under which rhabdomyolysis, muscle pain, or muscle weakness occur are also important to consider. FAO defects [21] and mitochondriopathies [22] are generally related to fasting or external precipitating factors (e.g., fever or superimposed illness), whereas high-intensity exercise (especially in sports) is more commonly responsible for symptoms in glycogenoses. Endurance sports are likely to trigger symptoms in FAO defects or mitochondrial disorders rather than in GSD [2].

Clues provided by the clinical history are sometimes unreliable or unreproducible, highlighting the continuum of clinical manifestations and the non-specificity of such indicators.

However, several clinical scenarios deserve particular attention from the clinician, as detailed below.

Rhabdomyolysis is an acute situation of immense muscle fiber damage during which serum CK levels rise ten times above normal values, putting the patient at risk for tubular necrosis, acute renal compromise, and possibly fatal multiorgan failure [23]. A level of 10,000 IU/l of creatine kinase is commonly accepted as a threshold for detecting myoglobin in the urine (myoglobinuria). Recurrent rhabdomyolysis may not be hereditary, nor metabolic, particularly in adults, but should always prompt a neuromuscular assessment including genetic testing [24].

Isolated hyperCKemia is a common biological finding often leading to a neuromuscular work-up. The extent to which this type of hyperCKemia should be explored is still debated. The diagnostic yield of NGS studies applied in this scenario has improved substantially during the last decade but remains below 50% in most cases [25].

Chronic myalgia represents a common complaint in the clinical setting and is not necessarily of metabolic origin. A minimal work-up including serum CK level dosage, EMG, and, in some, cases muscle MRI is nevertheless recommended.

Pseudometabolic presentation: Several genetically determined, non-metabolic myopathies can mimic a metabolic muscle disorder, such as Duchenne and Becker muscular dystrophies (DMD/BMD) and limb-girdle muscular dystrophies (LGMD), particularly sarcoglycanopathies and anoctaminopathies [26]. Myalgia, persistently elevated CK, and muscle pains/cramping are common in these etiologies.

Early multiorgan involvement is a rare presenting symptom that can be observed in some diseases, such as mitochondrial defects [27].

Asymptomatic patients: A rising number of patients with metabolic myopathies such as primary carnitine deficiency and infantile-onset Pompe disease are diagnosed after being screened in the neonatal period. Having an early diagnosis means they benefit from early treatment and may escape subsequent serious symptoms. Nevertheless, it is difficult to reliably predict the clinical outcome of some of these patients based only on biochemistry and genetic data.

An overview of the main clinical features of different types of metabolic myopathies is provided in Table 1.

### 2.5. Diagnostic Tools

Some investigations are basic and non-specific but remain helpful to document the overall clinical picture.

Serum creatine kinase (CK) is physiologically variable, even in healthy individuals. Normal serum CK values (100–300 IU/L according to local norms) do not preclude the diagnosis of metabolic myopathy. In contrast, it is not uncommon to come across very high figures (100,000 IU/L and above) in cases of severe rhabdomyolysis. Alongside serum creatine kinase levels, other laboratory features can help clinicians in diagnosis (Figure 3).

Nerve conduction studies and electromyography (EMG) are rarely very contributive in metabolic myopathies. A myogenic pattern or a normal pattern is the rule in overt myopathies. There are a few exceptions; for instance, a neurogenic component may be observed in some mitochondrial disorders.

The historical forearm ischemic test used to explore GSDs is usually replaced by a simple non-ischemic hand-grip test, during which simultaneous dosages of lactate and ammonia are administered sequentially before, during, and after exercise [28].

Muscle biopsies are becoming more sporadic but remain contributive in a few selected patients. PAS staining is particularly useful in GSD, and acid phosphatase histoenzymatic reactions are of interest in Pompe disease, while myophosphorylase can help detect myophosphorylase deficiency (McArdle disease) [29]. The presence of lipid droplets is a hallmark in many muscle lipidoses. More challenging to interpret is the presence of ragged red and/or COX-negative fibers that might be related to genuine primary mitochondrial dysfunction or simply to aging (personal observations). Ultrastructural studies are not essential for diagnosis, except to demonstrate the presence of polyglucosan bodies, active autophagy in typical Pompe disease, or paracrystalline inclusions in primary mitochondrial myopathies [29].

Nuclear magnetic resonance spectroscopy (NMR-S) applied to muscle is occasionally rewarding in FAOD and mitochondriopathies. It has been proven to be of limited interest or inconclusive in most other metabolic myopathies [30]. Moreover, this sophisticated investigation is not available everywhere.

NGS genetic testing for metabolic myopathies was introduced earlier in this review. In some centers and specifically in pediatric populations, trio analyses are proposed to enhance the diagnostic yield of such studies. In even more complex cases, RNA-seq and whole-genome sequencing (WGS) might provide additional clues. The investigation of mitochondriopathies is more challenging and hampered by the existence of two pools of genes contained either in the mitochondrial genome (mtDNA) or in the nuclear genome. To that end, NGS studies now include combined approaches to screen these genes more specifically (Mitome). Of note, a single mtDNA deletion is found only on muscle-derived mtDNA, especially in adults [31]. For this reason, in order to arrive at a certain diagnosis, a muscle biopsy is fundamental in these patients who do not transmit the single mtDNA to their offspring [31].

### 2.6. Diagnostic Algorithms

Diagnostic algorithms are based on simple, general principles.

The utmost priority is to identify a treatable or preventable disorder such as Pompe disease, primary carnitine deficiency (PCD), TK2 deficiency, or riboflavin-responsive MADD.

Early intervention is key to secure a clinical benefit for the patient, not only in terms of disease-modifying therapies (ERT and others) but also with respect to prophylactic measures (diets, training, and exercise protocols).

Should the condition be genetic, appropriate counselling must be provided within the family. Prenatal testing and preimplantation genetic diagnosis might be options in the most severe forms of early-onset metabolic myopathies.

A multidisciplinary approach is highly recommended, most preferably in reference centers dedicated to neuromuscular disorders or metabolic diseases.

Therapeutic interventions, whether specific or not, should ideally be evidence-based.

A thorough neurologic examination must be conducted in each patient. Except in situations of marked muscle weakness (GSD II, MADD, and others), the search for other clinical markers is often limited and not very rewarding. Checking systematically for other dysfunctions in other organs and tissues is logical (heart, liver, skin, brain, etc.), especially in mitochondrial disorders with muscle involvement.

In the most typical clinical presentations, direct genetic testing, whether targeted or not, is preferable. Since the vast majority of these disorders are autosomal recessive, sampling both parents is recommended in order to analyze any potential segregation of the pathogenic variants identified by NGS.

In challenging cases in which symptoms/signs are highly suggestive of metabolic myopathy but the NGS data are inconclusive, an extensive work-up should be considered, preferably in a specialized tertiary center. This would include, at a minimum, a muscle biopsy with a specific battery of stains, a hand-grip test, and various biochemical assays to investigate both glycogen and lipid metabolisms. Investigations should not be confined to muscle but involve other organs/systems (CNS, heart, lungs, liver, kidney, skin, etc.). Magnetic resonance spectroscopy (MRS) might also be useful in that context, despite its rather low diagnostic yield [30].

An asymptomatic individual with only mild hyperCKemia (<1000 IU/l) and a normal neurologic assessment may not need further investigations according to some authors.

The genetic nature of the condition can be inferred from a positive family history. The vast majority of metabolic myopathies are inherited following an autosomal recessive mode of inheritance. In that context, establishing a pedigree and highlighting potential consanguinity loops are useful clues. Beware of pseudodominant presentations in authentic recessive disorders when the kindred is part of an extremely inbred community. Other types of transmission have been described, for instance, x-linked inheritance including phosphorylase B kinase deficiency, phosphoglycerate kinase 1 deficiency, and complex-1 deficiency (mitochondrial), whereas autosomal dominant transmission remains an exception [32,33]. A significant number of mitochondrial myopathies are also maternally inherited. A simplified algorithm of the diagnostic approach to metabolic myopathies is presented in Figure 4.

Differential diagnoses encompass pseudometabolic myopathies (as described earlier), myopathies leading to episodes of severe rhabdomyolysis, and/or malignant hyperthermia but without a clearcut metabolic blockade such as *RYR1*-related and *CACNA1S*-related myopathies [34,35]. Non-hereditary autoimmune inflammatory myopathies (notably dermatomyositis) may also cause high CK, muscle edema, and pain in the context of a subacute onset. Acquired metabolic myopathies, although quite rare, always need to be considered (vitamin D deficiency, hypothyroidism, statin-induced immune necrotizing myopathy, etc.). This is why measuring vitamin D, TSH, and anti-HMGCR autoantibodies is important and should be part of the thorough metabolic work-up, especially in adults. Conversely, it is well-established that such external triggers may also reveal an underlying myopathy by lowering the threshold of clinical symptoms (especially rhabdomyolysis).

*Controversy.* For many years, myoadenylate deaminase deficiency was thought to be of metabolic nature. It appears more clearly nowadays that it represents a secondary, non-specific finding [36]. The so-called pathogenic variant NM_000036.3 c.34C>T;p.Gln12Ter in the *AMPD1* gene encoding myoadenylate deaminase presents more than 20,000 times in GnomAD and >1000 times in its homozygous state and is a common polymorphism in the general population. This deficiency is compensated by other metabolic pathways. Caution is therefore recommended in interpreting NGS data demonstrating variants in the *AMPD1* gene [36].

*Diagnostic tips.* A combination of muscular and non-muscular symptoms and signs may provide clues for the accurate diagnosis of a specific metabolic myopathy. For instance, the association between muscle lipidosis and ichthyosis is a cardinal feature of Chanarin-Dorfman syndrome (lipid storage disease with ichthyosis NLSD-I) [37]. Likewise, Jordans’ bodies are observed in blood smears in both NLSD-M and NLSD-I. Severe progressive cardiomyopathy is observerd in PCD, early-onset FAO defects, and, more rarely, LOPD. Encephaloneuromyopathies are mainly observed in mitochondrial diseases. The presence of moderate hyperuricemia (also called glycogenic hyperuricemia) points toward GDS V and VII (however, only in half of patients) [1]. Notably, this type of hyperuricemia rarely leads to bouts of gout. Deafness of variable severity and retinal impairment are sometimes observed in primary mitochondrial defects with multisystemic involvement. Hemolytic anemia is an indicator of GSD XII or, more rarely, GSD VII. Significant nausea and vomiting are common in mitochondriopathies and in GSD VII.

The time between the onset of symptoms and a diagnosis can be extremely variable; this type of heterogeneity depends on multiple factors, among which the most relevant are the clinical phenotype of the proband (some clinical presentations are highly specific, for example “second-wind phenomenon”/McArdle disease) and patient assessment at a reference center with expertise in neuromuscular diseases.

### 2.7. Therapeutics

Two types of interventions are to be considered: those meant to alleviate symptoms, notably muscle pain and/or exercise intolerance or, more importantly, to prevent the deleterious, sometimes life-threatening, consequences of severe rhabdomyolysis; and those specifically targeting a metabolic myopathy (ERT, nucleosides, multivitamin cocktails, etc.) [1].

Among the symptomatic interventions, guidance by a dietician plays an important role [38]. The way the patient and caregivers are capable of coping with exercise in general and with critical situations such as fasting and/or concomitant illness or stress is also of significance.

*Rhabdomyolysis* can be fatal if not properly treated. Intensive hyperhydration is necessary, and serum potassium and heart rhythm must constantly be monitored in an appropriate setting (intensive care unit if necessary) [24]. It is crucial to remain apprehensive of tubular necrosis, acute kidney failure, and cardiac arrest at every moment. Be wary if myoglobinuria is also absent when the patient presents with coma and anuria, as this is a red flag. Irrespective of the etiology, the mortality of rhabdomyolysis is around 2–5%, which is not negligible. With adequate management, CK levels decrease dramatically within 48 h in most cases. Checking the progressive normalization of CK levels after one week is critical. Persistent hyperCKemia beyond that date should prompt the clinician to undertake a comprehensive metabolic assessment and look for a hereditary etiology.

*Specific therapies* depend on the precise location and nature of the metabolic blockade. Enzyme replacement therapy is commonly used in GSD II, providing the missing GAA enzyme to muscle cells [4,5]. Similarly, the administration of nucleosides is meant to ameliorate the motor performance of patients with TK2 deficiency [39]. Pre-exercise oral sucrose administration has been shown to improve symptoms in McArdle disease [40]. Steroids given at the very beginning of CK elevation may be beneficial to avoid the often-fatal consequences of rhabdomyolysis in lipin-1 deficiency [41]. Bezafibrate [42] and triheptanoin have been used in FAOD with mitigated outcomes. Riboflavin supplementation often results in spectacular outcomes in patients with RR-MADD [43]. As for gene therapy in general, it remains in the preliminary stage. The efficacy and sustainability of pharmacological interventions in mitochondriopathies are debatable. Cocktails of vitamins associated with or without L-carnitine and CoQ10 can help [1].

Notably, very few disease-modifying therapies are evidence-based in metabolic myopathies. This is mainly due to the paucity of controlled therapeutic trials and to the limited number of patients enrolled, especially in ultra-rare conditions. Enzyme replacement therapy has nevertheless shown its efficacy in GSD II at the statistical level, leading to market approval by regulatory agencies (US Food and Drug Administration and the European Medicines Agency) [2].

## 3. Glycogen Storage Disorders

Glycogen storage disorders (GSDs) constitute a large family of hereditary conditions in which glycolysis or glycogenolysis can be impaired [19]. For the most part, they generate myopathic symptoms/signs, while a few of them are primarily hepatic in their clinical expression (GSD I, III, and IV). The other GSDs impair the glycogenolytic pathway mainly in muscle (GSD V, VII, and IX) or via glycolysis (GSD X, XI, XII, and XIII). However, recent studies suggest that the boundary between the two groups of GSD is not so clear. Pompe disease (GSD II) is traditionally regarded as a muscle glycogen storage disease, although some authors argue that it should not be so categorized, given its different underlying pathological mechanism, such as predominant autophagic activation.

Nonetheless, Pompe disease and McArdle disease represent the most frequent and the most paradigmatic forms of glycogen storage disorders that a treating neurologist or pediatrician is likely to encounter in their working lifetime.

### 3.1. Pompe Disease

Pompe disease (OMIM # 232300), also referred to as glycogen storage disease type II (GSD II), acid maltase deficiency, or α-glucosidase deficiency, was named after Johannes Pompe, a pathologist from the Netherlands who, in 1932, reported a postmortem case with a deleterious accumulation of glycogen in various tissues including skeletal muscle and heart [43]. This autosomal recessive condition was later found to be caused by pathogenic variants in the *GAA* gene (OMIM # 606800). The *GAA gene* encodes the α-1,4 -glucosidase (GAA), a lysosomal enzyme involved in the physiological degradation of glycogen molecules (glycogenolysis) [19]. GAA is not used to generate cellular energy for contracting myofibers, and it is not activated by physical activity. GAA is only used in lysosomes to break down glycogen that becomes engulfed within the lysosome during cellular autophagy. We note that from a purely biochemical point of view, Pompe disease is technically not a metabolic myopathy [44].

The prevalence of the disease is partially known and estimated to be around 1/40,000. The outcome of the various newborn screening programs in place in a couple of countries (Taiwan and USA) suggest that the overall prevalence might be significantly higher, especially in Asia [7].

The pathophysiology of Pompe disease has been extensively studied [43]. The vacuolization observed in many cells, preferentially those belonging to skeletal muscle and the heart, is due to dysfunctional lysosomes and autophagosomes. Autophagy is markedly increased as shown by various biomarkers. It is noteworthy that these deleterious cellular effects can also impact cells in the central nervous system, including motor neurons. The maximum threshold of residual GAA activity needed to develop symptoms is close to 30%. Some rare cases present with null GAA activity. This situation may constitute an obstacle to enzyme replacement therapies, given the immune intolerance it may generate.

Based on the age of onset, two forms of Pompe disease have been distinguished—infantile-onset Pompe disease (IOPD) and late-onset Pompe disease (LOPD)—the cut-off for which is arbitrarily and traditionally set according to the year of age [43]. However, in clinical practice, an entire continuum exists between these two subgroups. As IOPD presents with hypotonia and cardiomyopathy, it is not discussed in this review.

In LOPD, the spectrum of phenotypes is extremely wide, including childhood onset (after one year of age), juvenile onset, and adult onset. Symptoms and signs may be severe, leading to wheelchair confinement or assisted ventilation within a few years, and can sometimes be misleading (isolated ptosis, isolated respiratory insufficiency, non-specific limb-girdle weakness, chronic hyperCKemia, etc.).

LOPD manifests as a combination of muscle weakness and respiratory compromise. The diagnosis of Pompe disease is greatly facilitated by the use of a dried blood spot (DBS) taken in the consultation room [45]. Fluorometric DBS is a widely available first-tier investigation and should be widely prescribed in numerous clinical presentations (hypotonia, limb-girdle muscle weakness, myalgia, high CK, rigid spine, respiratory insufficiency, etc.). As the number of false-positive and false-negative cases remains high, a double check with a second test to measure GAA activity in other cells, tissues, or fluids (fibroblasts, muscle tissue, urine, etc.) is highly recommended.

A muscle biopsy, if performed, is likely to reveal a typical vacuolar myopathy characterized by multiple and variably sized vacuoles, making the HE pattern reminiscent of salami sausages. The vacuoles are filled with PAS-positive material, and some of these are intensely stained with acid phosphatase, a reliable biomarker of autophagy [29]. In LOPD, vacuolization is far less prominent and can even be absent in roughly thirty percent of muscle biopsies.

Genetic testing has become the investigation of choice to confirm the diagnosis of Pompe disease once suspected on the basis of biochemical clues (DBS and other biochemical assays) [19]. More than 2000 variants of the *GAA* gene, half of which are disease-causing, are known and enlisted in the international Pompe registry [46], corresponding to 1079 patients. The intronic splicing mutation c.-32-13T>G (also called IVS1) is the most common, especially in Caucasian patients. In French Guiana, a founder effect originating from West Africa has been identified [9]. GAA pseudodeficiency has been reported on numerous occasions, especially in Asia, and corresponds to specific alleles.

Patients with only a single pathogenic mutation and a compatible phenotype are not uncommon, often leading to a therapeutic dilemma. In such cases, a muscle biopsy can be justified to validate either a variant of unknown significance (VUS) in the *GAA* gene or within the framework of a therapeutic trial. GAA enzyme activity can also be measured in muscle, but this assay is not available in all centers [19].

Other investigations of Pompe disease are not confirmatory but only indicative. Serum CK levels are variably elevated at diagnosis but may be mildly elevated or in the normal range at the advanced stage of the disease due to reduced muscular tissue. Electrophysiological studies may be useful for the differential diagnosis but are rarely conclusive. Reports of neurogenic patterns on ENMG represent an exception to the generally observed myopathic pattern. Clinicians should also be aware of the pseudomyotonic discharges often found in paravertebral muscles of patients, which are possibly correlated with the degree of muscle vacuolization. Muscle imaging might be useful, although not particularly so for diagnostic purposes. The relatively high prevalence of usually asymptomatic aneurysms in cerebral vessels may be (systematically for some teams) an indication for brain imaging. Such imaging studies may also show asymptomatic white matter changes in some treated or untreated patients.

ERT for LOPD patients is administered intravenously every two weeks and sometimes at home in some countries. Its efficacy is highly debated despite convincing data gathered from several clinical trials. The patient actually benefits in terms of muscle fatigue, muscle performance (measured by the six-minute walking test), and respiratory autonomy. ERT has been available for patients with LOPD since 2010 [4].

Treating individuals with molecularly proven GSD II based solely on simple myalgia with no additional muscle weakness remains controversial. Efforts have been made at the European level to reach some consensus in these challenging situations.

Long-term administration of first-generation ERT has its limitations. Many patients respond less or stop responding altogether to supplementation, let alone the burden imposed on the patient, the caregivers, and the payers. This pharmacological intervention is administered intravenously twice a month and at home if possible. A new generation of ERT products has been developed, either alone (the neorecombinant GAA manufactured by Sanofi-Genzyme) or in combination with a chaperone molecule called miglustat (Amicus). The two investigational studies with these novel compounds (COMET and PROPEL respectively) showed encouraging results in terms of efficacy, but their real added value when compared to conventional ERT is statistically debatable [5]. Such approaches may nevertheless be approved as a second-tier therapy in conventional ERT-resistant patients.

In parallel, other biopharmaceutical work on alternative approaches such as gene therapy, oligo antisense nucleotides, or gene editing targets the *GAA* gene itself. Gene replacement therapy (GRT) represents a real opportunity to treat GSD II more efficiently. Some biopharmaceutical companies recently applied distinct strategies in the clinical setting—some with GRT products targeting the liver and others targeting the muscle itself. The low rate of patients naive to AAVs and the inability to dose the patient more than once constitute significant obstacles. Safety concerns have also been raised in some participants, particularly immune reactions during the clinical development of GRT, as also encountered in other trials probing GRT [4].

Many patients with PD show evidence of sarcopenic obesity with low muscle mass and a high body fat percentage. Exercise and nutrition are considered prominent complementary therapeutic strategies. To optimize muscle protein synthesis, a diet rich in high-quality protein (for example, milk and eggs) is suggested. Specific amino acid supplementation seems to be another therapeutic perspective, especially L-leucine supplementation, as it appears to be the most powerful in stimulating protein synthesis through mTOR activation [47,48].

Exercise was found to be beneficial in that it enhanced autophagy and mitochondrial biogenesis. [48] In particular, a moderate-intensity aerobic and strength exercise training program in LOPD patients receiving ERT appears to have a beneficial effect in terms of slowing the loss of muscle function [49].

### 3.2. McArdle Disease

McArdle disease or GSD V (OMIM #608455) is a glycogen storage disorder caused by mutations in the *PYGM* gene encoding an enzyme called myophosphorylase [1]. Glycogen molecules are no longer catalyzed in glucose-6-phosphate and accumulate at the subsarcolemmal level of the muscle fiber.

Clinically, the disease leads to exercise-induced cramping and myalgia, concomitant elevation of CK, and post-exercise myoglobinuria (dark-colored urine, Coca-Cola port wine-colored urine); extramuscular manifestations are extremely rare. Symptoms are usually observed in childhood and adolescence. Patients are reportedly not good at sports. The second-wind phenomenon is prominent and may represent a key to diagnosis. Patients develop symptoms consisting of myalgia and pain within one minute of high-intensity effort and have to stop or reduce the workload to a lower level. After about 10 min, they can experience the second-wind phenomenon, with a decrease in HR and an improvement in exercise tolerance. This virtually pathognomonic phenomenon corresponds to a switch in metabolic pathways (from carbohydrates to lipids) but may be absent in some individuals with GSD V. Respiratory compromise remains an exception. Episodes of secondary gout have been reported occasionally without any synchrony with bouts of rhabdomyolysis [1]. Extramuscular manifestations such as ptosis and retinal dystrophy are exceptionally reported in GSD V.

McArdle disease is probably the most common subtype of muscle glycogenoses, with an estimated prevalence of between 1/67,000 and 1/100,000 inhabitants. Various initiatives have been launched to promote a census of patients with GSD V, including a European Registry for McArdle disease [50].

The mode of inheritance of McArdle disease is invariably autosomal recessive. One pathogenic variant (Arg50*) found in the homozygous or heterozygous state accounts for 60% of cases, at least in Caucasian populations. Autosomal dominant transmission has been documented in one family in which patients presented adult-onset muscle weakness without exercise intolerance that also differed from McArdle cases at the metabolic level [51]. Muscle biopsies showed PAS-positive inclusions, and electron microscopy showed that the latter consisted of highly ordered honeycomb structures surrounding glycogen-free areas in the muscle [51].

The diagnosis of McArdle disease can be strongly suspected at the time of clinical history and confirmed by ancillary tests. Interictal neurologic examination is normal except in patients who exhibit permanent muscle weakness, often after 40 years of age. Marked CK elevation is a cardinal feature during exercise-induced myalgia (see rhabdomyolysis). Creatine kinase seldom normalizes totally and remains mildly increased (two to fivefold normal values). The forearm ischemic test tends to be replaced by less invasive investigations [28]. In typical cases, lactate curves remain flat post exercise, while ammonia levels remain high. Muscle histology, when performed, shows a vacuolar myopathy of variable intensity with PAS-positive material accumulation combined with a negative myophosphorylase histoenzymatic reaction [29]. The identification of mutations in the *PYGM* gene located on chromosome 11 is confirmatory and can be achieved either by Sanger sequencing (of the recurrent Arg50* or, if negative, of the whole *PYGM* gene) or by NGS [19].

The differential diagnosis of McArdle disease is that of metabolic myopathies in general and of other GSDs, in particular Tarui disease (GSD VII or phosphofructokinase (PFK) deficiency), which can mimic GSD V without the second-wind phenomenon [52].

The functional prognosis of McArdle disease is fairly good, provided the patient manages and copes with episodes of rhabdomyolysis that inevitably occur in their lifetime and that may otherwise result in serious complications. In the long run, some patients present with some degree of persistent residual muscle weakness due to increasing vacuolization within muscle fibers [52].

McArdle patients require a global approach that integrates nutritional aspects, supplementation, and exercise.

Different types of dietary approaches have been trialed. It seems that a diet rich in carbohydrates has more beneficial effects in terms of a significant drop in the heart rate, maximal work capacity, and exercise tolerance compared to a diet rich in protein [53].

The low-carbohydrate ketogenic diet (LCKD), according to patient-reported experiences, presents, in 90% of cases, some degree of positive effect, in particular on activity intolerance, muscle pain, and muscle fatigue. Clinical trials are required to examine the therapeutic potential of a LCKD [54].

Low-dose creatine supplements seems to have minor benefits in terms of improving exercise tolerance in a small number of people with the condition [55]. Regarding physical exercise, regular aerobic exercise may lead to physiological adaptations that increase oxidative and work capacity in patients with McArdle disease [49].

### 3.3. Other Muscle Glycogenoses

The other GSDs are far less common.

GSD III (Cori–Forbes disease or glycogen debranching enzyme deficiency or OMIM #232400) is the third most prevalent muscle glycogenosis after GSD II and GSD V. Clinically, GSD III is a biphasic disorder. During childhood, patients present with hepatomegaly and severe fasting hypoglycemia, hyperlipidemia, and hyperketonemia. All these situations are highly suggestive of liver metabolic dysfunction. Later in life, in adolescence or adulthood, patients develop a progressive myopathy that can be accompanied by muscle weakness and exercise intolerance [56]. During this phase, the metabolic impairment is less prominent, and the patients are referred to a neurologist. A minor percentage (15%) of patients develop cardiomyopathy and other liver complications, such as cirrhosis. Hepatocellular adenomas (HCAs) and carcinomas (HCCs) have previously been described in GSD III [57]. The diagnosis relies on the identification, by NGS, of biallelic mutations in the *AGL* gene. 

Dietary management tailored to the individual patient remains the primary therapy. In infancy, frequent feeding (every 3–4 h) is needed to maintain euglycemia. A diet rich in protein intake (3 g/kg) is recommended; on the other hand, a high-fat diet can be considered to reduce cardiomyopathy. Intake of maltodextrin or rapidly absorbable carbohydrates prior to exercise can prevent hypoglycemia during physical activity, and oral fructose and sucrose ingestion can improve exercise tolerance [58].

A gene replacement approach has been developed in a mouse model of GSD III. Secondary activation of autophagy has recently been shown in muscle biopsies from GSD III patients, also opening novel therapeutic avenues [59].

Glycogen branching enzyme (GBE1) is one of the three main enzymes responsible for glycogen synthesis. Mutations in the *GBE1* gene have been associated with GSD IV. The neuromuscular form of GSD IV, which was probably underestimated for a long time, has a wide range of presentations with variable onset and severity [60]. Three clinical forms based on age at onset have been described in GSD IV. The first comprises cases with antenatal onset, polyhydramnios, decreased fetal movements, fetal akinesia deformation sequence (FADS), arthrogryposis multiplex congenita (AMC), hydrops fetalis, and perinatal death [61]. The second presents with severe congenital myopathy, often mimicking spinal muscular atrophy [62]. The third and most benign myopathic form is characterized either by delayed motor milestones, proximal muscle weakness, or proximal muscle weakness manifesting in adulthood [63].

### 3.4. Polyglucosan Body Myopathies

Polyglucosan bodies result from the accumulation of amylopectin-like polysaccharides in both skeletal and cardiac muscle tissues. They lead to myopathic manifestations such as muscle weakness and wasting, cardiomyopathy with arrhythmia, conduction blocks, and cardiac failure. In addition to excessive storage in muscle, polyglucosan bodies can affect other tissues, such as the brain, causing CNS-related symptoms, often as the main manifestation (notable in adult polyglucosan body disease) [64].

*Glycogenin-1*-associated disorders include GSD XV, manifesting as a pure cardiomyopathy with myocardial polyglucosan bodies with a non-functional glycogenin-1, and polyglucosan body myopathy 2 (PGBM2), presenting with a pure skeletal myopathy with variable distribution of muscular weakness, with a limb-girdle phenotype or distal myopathy [64]. Muscle biopsies from PGBM2 patients show accumulation of glycogen and polyglucosan bodies, as well as depletion of glycogenin-1 [65].

*RBCK1 (HOIL-1)* deficiency can manifest either as an autosomal recessive lethal immunodeficiency with autoinflammation and polyglucosan accumulation in muscle, heart, and liver or as polyglucosan body myopathy type 1 (PGM) with severe dilated cardiomyopathy and polyglucosan bodies found in skeletal muscle and heart [64]. These conditions, although extremely rare, can provoke a life-threatening cardiac compromise. A muscle biopsy is recommended to confirm the presence of polyglucosan bodies after the identification of variants in these genes by NGS.

For a detailed description of these entities, please refer to [66].

## 4. Lipid Myopathies

### 4.1. Fatty Acid Oxidation Disorders

Fatty acid oxidation (FAO) disorders are inherited autosomal recessive diseases caused by a deficiency of enzymes involved in the metabolism of lipids. The absence of these key enzymes leads to metabolic dysfunction with accumulation of lipid droplets (LDs) in several organs (skin, liver, heart, and particularly in skeletal muscle). Due to their common pathogenesis and histological features, they are also defined as lipid storage myopathies (LSMs) [2]. Such myopathies include primary carnitine deficiency (PCD), multiple acyl-coenzyme A dehydrogenase deficiency (MADD), neutral lipid storage disease with ichthyosis (NLSD-I), and neutral lipid storage disease with myopathy (NLSD-M). Carnitine palmitoyl transferase deficiency (CPT II) traditionally falls into the group of LSMs, despite the absence of lipid accumulation in this entity.

Fatty acids represent an essential element for energy homeostasis in cardiac and skeletal muscle; in some conditions, such as fasting or prolonged moderate exercise, muscle cells strictly depend on FAO to develop energy. Fatty acids are classified according to their size into short-chain (SCFA), medium-chain (MCFA), long-chain (LCFA), and very-long-chain (VLCFA) fatty acids [1]. Among these, LCFAs (the largest circulating fraction) are the main substrate of lipid metabolism in cells. While SCFAs and MCFAs freely diffuse across the membranes, LCFAs and VLCFAs, because of their size, require specific transporters to cross mitochondrial membranes before being involved in β-oxidation [67,68].

FAO defects cause a progressive accumulation of non-degraded lipids in muscle cells. This leads to a symptomatology-mixing myalgia, exercise intolerance, and episodes of massive rhabdomyolysis with myoglobinuria and kidney failure. Interestingly, and as opposed to glycogen storage disorders, acute episodes are not often triggered by high-intensity muscle exercise but various other factors, such as fever with or without concomitant infection, fasting, sustained low-intensity exercise, emotional stress, or cold-induced thermogenesis. Some laboratory clues can help clinicians to suspect FAOD [19], such as hypoketotic hypoglycemia (due to the rapid metabolization of glucose); an increase in the free fatty acids (FFAs)/ketones ratio in serum (from 1:1 to greater than 2:1), suggesting a β-oxidation block and modifications of various biological parameters such as the serum levels of total and free plasma carnitines, urine, and plasma acylcarnitines; and urinary organic acids.

#### 4.1.1. Primary Carnitine Deficiency

Primary carnitine deficiency (PCD) is a rare metabolic disorder characterized by low concentrations of carnitine in plasma and tissues such as the liver, kidney, skin, heart, and skeletal muscle. PCD was reported for the first time in 1973 and has been documented in more than 750 patients in literature [69]. However, a meta-analysis of the literature recently showed that a significant number of those reports were actually cases of MCAD deficiency or secondary carnitine deficiency [69].

PCD is traditionally categorized into two different forms: (1) muscle carnitine deficiency (OMIM #212160), which is characterized by lipid storage myopathy, predominantly proximal limb and neck muscle weakness, with low levels of muscle carnitine but normal carnitine levels in liver and plasma; or (2) systemic carnitine deficiency (OMIM #212140) with low carnitine in the plasma and liver only [2].

Systemic carnitine deficiency is an autosomal recessive disease caused by homozygous or compound heterozygous mutations in the solute carrier family 22 member 5 gene (*SLC22A5*), which encodes the organic cation/carnitine transporter 2 (OCTN2). OCTN2 is localized on the cell membrane, and its function is to actively transfer carnitine from the plasma to cytosol [21].

Secondary carnitine deficiency, which should be distinguished from the abovementioned primary carnitine deficiency, may be encountered in several systemic conditions, such as organic aciduria, malnutrition, Fanconi syndrome, chronic hemodialysis, or drug-induced toxicity (zidovudine and valproate in particular). In PCD, muscle carnitine levels are significantly lower (<2% to 4% normal values) compared with moderately raised levels observed in secondary carnitine deficiency (25–50% normal values) [70].

PCD has various clinical presentations in terms of organ involvement, severity, and age of onset. The disease commonly appears in childhood and is characterized by bouts of rhabdomyolysis and hypoketotic hypoglycemia, generalized muscle weakness, and Reye-like syndrome. Young patients often suffer from hypertrophic or dilated early-onset cardiomyopathy. In adult-onset forms, muscle fatigability and cardiac arrhythmias are the usual presenting symptoms. On the whole, cardiomyopathy is the most prevalent symptom, with an unpredictable risk of sudden death [2,21,69].

In the systemic form of PCD, serum CK levels can be variable, from normal to 15 times normal values, while liver enzymes (ALAT and ASAT) can be elevated [71]. EMG images may also be normal or favor a myopathic process. Hypertrophic or dilated cardiomyopathy may be an early symptom detectable via echocardiography. Muscle histology often shows vacuoles, as well as a massive subsarcolemmal and intermyofibrillar lipid accumulation; type I fibers are mostly affected. Testing of the *SLC22A5* gene is diagnostic, while functional studies of the reduced carnitine uptake in lymphocytes or fibroblasts are helpful in inconclusive cases. A total of 150 pathogenic variants of the *SCL22A5* gene are currently listed in various databases, the majority of which are missense mutations [21,70]. Sequence analysis is clinically available and can detect at least one mutation in approximately 70% of affected individuals. Large deletions and duplications of the *SLC22A5* gene, while not a common mechanism causing systemic carnitine deficiency, have been identified in at least one individual with this diagnosis. Therefore, if sequence analysis is negative or only detects one mutation in an individual strongly suspected of having systemic carnitine deficiency, then array comparative genomic hybridization (aCGH) is recommended to screen for larger deletions and duplications that may involve part of or the entire *SLC22A5* gene [72].

The main treatment for PCD is based on oral supplementation of high-dose carnitine (from 100 to 600 mg/kg/day) in four daily doses calculated based on carnitine depletion from tissues. Restoration of the L-carnitine level greatly improves cardiomyopathy and myopathy and, if administered during the first weeks of life, can prevent the development of the phenotype in newborns [2,21].

National screening programs, or simply pilot studies, meant to detect PCD at birth have been carried out or implemented in various countries since 2001. Taiwan, a couple of places in mainland China, Japan, and Australia are particularly active in that respect. Such newborn screening programs based on mass-tandem spectrometry enable, in theory, early introduction of carnitine supplementation and prevention of serious medical complications in the long run. However, in real-world conditions, the screening of PCD deficiency seems more complex, and the decision to treat it prophylactically is more difficult to make. Multiple hurdles have appeared in the process of NBS for PCD. False-positive cases are numerous. Low levels of serum carnitine in a newborn do not necessarily imply that the proband has PCD but might be the result of a passive transfer by a hitherto asymptomatic mother. It is well established that a considerable number of individuals (up to 50%) harbor pathogenic variants of the *SLC22A5* gene but remain asymptomatic throughout life. Genetic testing (segregation analysis of the *SLC22A5* variants), a subsequent dosage of carnitine after one week, and functional studies (to assess carnitine transport activity) help sort this out. These theoretical and/or practical difficulties have led many countries to not implement NBS or to abandon such programs, such as in New Zealand.

#### 4.1.2. Multiple Acyl-CoA Dehydrogenase Deficiency

Multiple acyl-CoA dehydrogenase deficiency (MADD) or glutaric aciduria type II (GA II) (OMIM #231680) is an autosomal recessive disorder of fatty acid and amino acid metabolism. MADD is caused either by mutations in the *ETFA* and *ETFB* genes encoding one of the two subunits of the electron transfer flavoprotein or by mutations in the *ETFDH* gene that encodes the electron transfer flavoprotein dehydrogenase (ETFDH). The latter is an essential protein in the catalysis of the initial step of mitochondrial fatty acid β-oxidation [68]. Clinically, MADD is subdivided into three main forms: type I, neonatal onset with congenital anomalies; type II, neonatal onset without anomalies; and type III, mild or late onset. Patients with type I or II MADD present a severe form characterized by neonatal onset and death in the first weeks or months despite treatment. Metabolic acidosis frequently associated with hepatomegaly, hyperammonemia, and profound hypoglycemia (often within the first 24–48 h of life), represent the main clinical manifestations. Affected newborns are often premature and hypotonic and may have a characteristic smell (sweaty feet-like) [21]. The congenital anomalies commonly found in type I MADD are facial dysmorphism, cystic kidneys, macrocephaly, pachygyria, and cortical heterotopias. In patients surviving the neonatal period, hypertrophic cardiomyopathy and recurrent episodes of metabolic decompensation are typical of the syndrome.

Type III MADD is a late-onset lipid storage disease caused by mutations in the *ETFDH* gene and the most common form of MADD. The clinical spectrum is wide, from exercise intolerance and muscle pain to muscle weakness with proximal myopathy and rhabdomyolysis. The phenotype observed in late-onset MADD patients is also called riboflavin-responsive MADD (RR-MADD) [2], since clear clinical improvement is noted following high-dose riboflavin supplementation. Mutations in the *ETFDH* gene lead to a secondary CoQ10 deficiency [73]. This finding justifies documenting each case with muscle histology; besides reversible massive lipid droplet accumulation, COX-negative fibers and ragged red fibers (RRF) can also be found in abundance [21]. The late-onset form of MADD due to the *ETFDH* gene is particularly frequent in Far East Asia (China, Japan, and Korea).

In patients with MADD, a diet poor in proteins and fats and avoidance of long fasting periods can be helpful. In RR-MADD patients, supplementation therapy includes high-dose riboflavin (50–100 mg 2–3 times daily), coenzyme Q_10_ supplements (60–240 mg daily in two divided doses), and carnitine supplementation in those with deficiency [74].

#### 4.1.3. Carnitine Palmitoyl Transferase II Deficiency

Carnitine palmitoyl transferase II deficiency (CPT II deficiency, OMIM #255110) is an autosomal recessive long-chain fatty acid oxidation (LCFA) disorder caused by the inability of LCFA, the major FFA circulating fraction, to spontaneously move through mitochondrial membranes. Three different clinical presentations of CPT II deficiency are known: (1) a lethal neonatal form, (2) a severe infantile hepatocardiomuscular form, and (3) a mild myopathic form with onset in childhood or adulthood.

Muscle CPT II deficiency, caused by mutations in the *CPT2* gene, represents the commonest form of lipid metabolism disorders found in adults [68]. The CPT system is based on the activity of two distinct enzymes: CPT I, located in the outer mitochondrial membrane, and CPT II, localized in the inner mitochondrial membrane, which catalyzes the transesterification of palmitoyl carnitine back into palmitoyl-CoA [2].

Clinical manifestations of muscle CPT II deficiency are characterized by episodes of myalgia, transient weakness, and, in the most acute forms, severe bouts of rhabdomyolysis (CK 50.000–200.000 UI) with myoglobinuria. This disorder is observed in young adults exposed to specific triggers such as fasting, prolonged moderate exercise, a high-fat diet, cold exposure, anesthesia, emotional stress, or fever. Between attacks, clinical evaluation, CK levels, and EMG are generally normal; given these features, muscular CPT II deficiency can be defined as an intermittent induced muscular necrosis. Interestingly, CPT II deficiency is characterized by the absence of accumulation of intramuscular lipid droplets in muscle fibers [67]. Due to the non-specificity of many of these clinical presentations, differential diagnoses often have to be considered, such as McArdle disease (GSD V), in which the typical “second-wind” phenomenon is typically observed.

#### 4.1.4. Neutral Lipid Storage Diseases

Chanarin–Dorfman syndrome (CDS; OMIM #275630), also referred to as neutral lipid storage disease with ichthyosis (NLSD-I), is an inborn error of lipid metabolism caused by mutations of the *ABDH5* gene [37]. CDS is an ultra-rare autosomal recessive disorder, the majority of cases having been described in the Middle East, Turkey, the Indian subcontinent, and the Mediterranean basin, where endogamy is high [37]. To date, a total of 150 cases of CDS have been identified. *ABDH5* (also named *CGI-58*) encodes an activator of the adipose triglyceride lipase involved in the first steps of fatty acid oxidation. The clinical hallmark of this condition is congenital-onset ichthyosis of variable intensity. Muscle weakness, hepatosplenomegaly, ocular abnormalities (cataracts and ectropion), and sensorineural deafness are accompanying symptoms. Muscle manifestations occur rather late (in the thirties, on average) and can be seriously debilitating. Liver involvement may lead to hepatic failure and even cirrhosis over time.

The association of ichthyosis and myopathy is highly suggestive of CDS and may be corroborated by the presence of marked lipid droplets in muscle fibers (if muscle histology is documented), as well as in granulocytes (so-called Jordans’ bodies). Besides this classic presentation, it is likely that milder forms of CDS exist without any myopathic signs, thereby expanding the phenotypic spectrum. A low-fatty-acid diet in combination with vitamin E and ursodeoxycholic acid may be helpful in some patients, in addition to symptomatic interventions meant to alleviate the consequences of ichthyosis.

Neutral lipid storage disease with myopathy (NLSD-M, OMIM #610717) is a rare autosomal recessive disorder caused by mutations in the *PNPLA2* gene that maps to chromosome 11, reducing normal expression or function of the ATGL protein and leading to the accumulation of triglycerides in various tissues [75]. Clinically, NLSD-M results in muscle manifestations, including cardiomyopathy, more commonly in adulthood than in childhood. Muscle weakness is often triggered by exercise, fasting, or infections. Interestingly, individuals with NLSD are not typically obese. It has been proposed that the assimilation rather than degradation of triglycerides is the main factor in fat accumulation in adipose cells. Muscle histochemistry, if performed, shows marked features of muscle lipidosis, and blood smear may also demonstrate some microdroplets in leukocytes.

## 5. Primary Mitochondrial Myopathies

Numerous mitochondrial cytopathies have a myopathic component as part of their clinical expression [3]. Ophthalmoparesis, ptosis, generalized weakness, fatigue, exercise intolerance, and rhabdomyolysis are also common presenting symptoms. In that sense, mitochondrial myopathies overlap with the above-mentioned lipid myopathies, and the majority of are characterized by the presence of abundant ragged red and COX-negative muscle fibers, as well as lipid accumulation in muscle [29]. Another frequent trait of mitochondriopathies is the concomitant impairment of multiple tissues: heart, brain, liver, bone marrow, nerve, etc. Some mitochondrial cytopathies such as MELAS (mitochondrial encephalopathy, lactic acidosis, and stroke-like episodes), MERRF (myoclonic epilepsy with ragged red fibers), and Kearns–Sayre syndrome are good examples of this phenotypic diversity. Chronic progressive external ophthalmoplegia (CPEO) is also commonly a presenting sign. Maternal inheritance is an indicator of mitochondrial disorder, but other modes of transmission are also possible (X-linked, autosomal dominant, and autosomal recessive). Biologically, the rise in serum lactate is an interesting clue, despite numerous false positives. NGS studies allow for simultaneous screening of both mitochondrial and nuclear genes associated with mitochondrial diseases. Novel mitochondrial genes of nuclear origin are regularly discovered.

Dietary supplements are commonly used in clinical practice for patients with mitochondrial myopathies, although little evidence of safety or efficacy from well-designed clinical trials exists.

The five most frequently used supplements are CoQ_10_ (28%), L-carnitine (25%), vitamin D (19%), riboflavin (16%), and vitamin C (12%). CoQ_10_ is the most widely used supplement, and it should be administered to most patients with a diagnosis of mitochondrial disease and not exclusively for primary CoQ_10_ deficiency at the standard dose of 2 to 8 mg/kg per day twice a day with food [76,77].

These supplements can be variously mixed a the so-called “mito cocktail” a highly individualized supplement mix that can includes coenzyme Q10, α-lipoic acid, vitamin E, creatine monohydrate, folate, vitamin B12, vitamin D, and riboflavin [1]. Exercise is another important therapeutic approach in patients with mitochondrial myopathies; in particular, aerobic training can efficiently improve oxidative capacity, with beneficial effects on fatigue and tolerance to daily activities [78].

Among all the mitochondrial myopathies, one in particular needs to be highlighted, as it is treatable: TK2 deficiency [39].

### 5.1. TK2 Deficiency

Thymidine kinase type 2 (TK2) is a nuclear-encoded mitochondrial enzyme involved in the metabolism of nucleosides. It catalyzes the first phosphorylation of pyrimidine deoxynucleosides taking place in the mitochondrion. Its impairment leads to an autosomal recessive mitochondrial depletion syndrome (MDS) in both children and adults and triggers several and variably severe manifestations, including myopathy. The condition, which is certainly underdiagnosed, remains ultra-rare, with only 100 cases reported in the literature or in databases to date. Hispanic ethnicity might be a predisposing factor according to [12].

Cardinal features of the disease include muscular weakness, CK elevation, the presence of ragged red muscular fibers, high lactate levels in some cases, low mitochondrial DNA copy number in muscle tissue, and a faulty respiratory chain. The prognosis of TK2 deficiency in children is extremely poor, with dramatically rapid disease progression. The early-onset form of the disease is particularly devastating, with a mortality close to that of spinal muscular type I before the arrival of disease-modifying therapies. In adults, muscle impairment occurs late but may also be devastating, leading to the loss of ambulation and/or assisted ventilation. The unusual association of dystrophic features (necrosis and regeneration and fat replacement) and numerous ragged red fibers and/or COX-negative fibers in muscle are highly suggestive [79]. Depletion of the mitochondrial membrane is often observed in the most severe forms of TK2 deficiency. The clinical presentation may be misleading; some infants or children have been misdiagnosed with SMA; diagnosis is even more challenging in the few infants presenting with encephalomyopathy. In juvenile/adult-onset forms, chronic progressive external ophthalmoplegia is sometimes the first and unique complaint for many years. Generally speaking, later-onset forms of TK2 deficiency can mimic a wide spectrum of neuromuscular disorders, including facioscapulohumeral dystrophy (FSHD), oculopharyngeal muscular dystrophy (OPMD), LOPD, and even SMA type 3 [12]. The wider use of WES allows us to diagnose this entity, and, in the near future, *TK2* mutations are expected to be discovered in children, mostly infants, screened by whole-exome sequencing in the context of severe hypotonia, and new phenotypes may emerge. The same trend is likely to occur in adults.

Early recognition of TK2 deficiency is key to undertaking a specific treatment based on the oral administration of nucleosides [39]. The clinical outcome can be remarkable in infants when treated soon after initiation of symptoms. An open-label, compassionate-use study with deoxynucleoside monophosphates and deoxynucleotide seems very promising and may pave the way toward further approval of this pharmacological, rather cheap intervention by regulatory agencies. Early-onset forms benefit the most from these oral supplementations in nucleosides. Some patients have even been weaned off their ventilators. A phase II trial (NCT03845712) is ongoing (https://www.clinicaltrials.gov/ct2/show/NCT03845712, accessed on 4 April 2023). However, one obstacle is that oral nucleosides are not regarded as medicinal products, impeding their use and dissemination.

### 5.2. Lipin-1 Deficiency

Lipin-1 deficiency (phosphatidic acid phosphatase deficiency) has emerged in the last two decades as a major cause of autosomal recessive fever-induced rhabdomyolysis in young children [80]. Lipin-1 is a protein involved in the PPAR-γ pathway, leading to an impairment in lipid biosynthesis. If untreated and/or detected late, complications are potentially lethal. Exercise, fasting, and anesthesia are far less common triggering factors. Cardiac arrest has been documented in numerous observations, irrespective of any bout of rhabdomyolysis, suggesting the existence of a direct, potentially lethal cardiotoxicity (cardiac arrhythmia). Lipin-1 has also been described in adulthood but with less severe presentations and a lower vital risk. Between episodes of rhabdomyolysis, there are no clinical or biological clues to guide clinicians. Acylcarnitine profiles, as well as urine organic acid chromatography, are unremarkable. Lipidosis has not been reported in muscle fibers either. To date, the only confirmatory test is of genetic nature (using metabolic or rhabdomyolysis gene panels or clinical exome sequencing). One variant is particularly frequent in Caucasian patients. Symptomatic treatment of rhabdomyolysis is key, and recent data suggest that early administration of steroids at the initiation of the crisis might also be beneficial [81]. Prevention of rhabdomyolysis is also possible by educating caregivers to detect at-risk scenarios and by admitting the child to ICU for monitoring and intervention.

## 6. Conclusions

The landscape of metabolic myopathies has significantly changed over the last two decades. Diagnostic algorithms have integrated the need for first-tier genetic testing, especially when the clinical picture is atypical, unspecific, and/or incomplete. In parallel, an ongoing therapeutic revolution is addressing an increasing number of metabolic myopathies, especially the most severe ones. In that respect, gene therapy is promising, although still developing. We are hopeful that simple, cost-effective, and widely available medications (riboflavin) are already capable of rescuing a few phenotypes. One should not forget that management of daily activities including physical exercise and dietary habits contributes to the alleviation of symptoms and can prevent complications such as rhabdomyolysis.

From a nosological standpoint, it might be time and of interest to reappraise the current classification of metabolic myopathies. Stricto sensu, the term encompasses muscle disorders involved in impairment of energy supply in one way or another. Interestingly, an increasing number of defective metabolic pathways have been found to be involved in muscle disorders (i.e., GNE myopathy and hyposialylation) and, more generally, in neuromuscular disorders (SORD2-related Charcot–Marie–Tooth disease). Pooling this in a more comprehensive entity called neuromuscular metabolic disorders might make sense.

## Figures and Tables

**Figure 1 genes-14-00954-f001:**
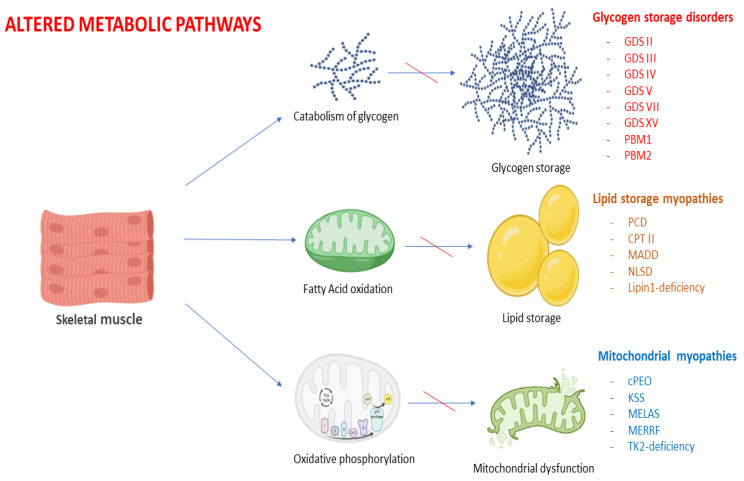
**Metabolic pathways and corresponding metabolic myopahties.** Created with BioRender.com (https://www.biorender.com/ accessed on 4 April 2023).

**Figure 2 genes-14-00954-f002:**
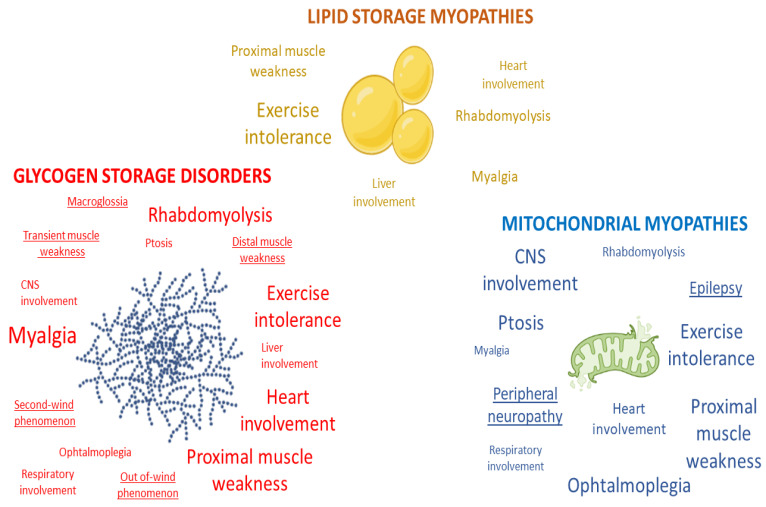
Clinical symptoms and/or signs highly suggestive of metabolic myopathies. Font size depends on the frequency of the element in the specific group. The highlighted features are found only in the group they belong to (created with BioRender.com, https://www.biorender.com/ accessed on 4 April 2023).

**Figure 3 genes-14-00954-f003:**
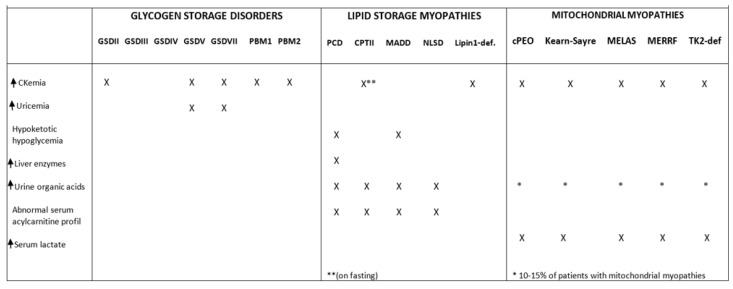
**Biomarkers in metabolic myopathies.** “ ** ” in patients with CPT II, hyperCKemia is frequent during fasting. “ * ” 10–15% of patient with mitochondrial myopathies can present abnormalities in urine organic acids level.

**Figure 4 genes-14-00954-f004:**
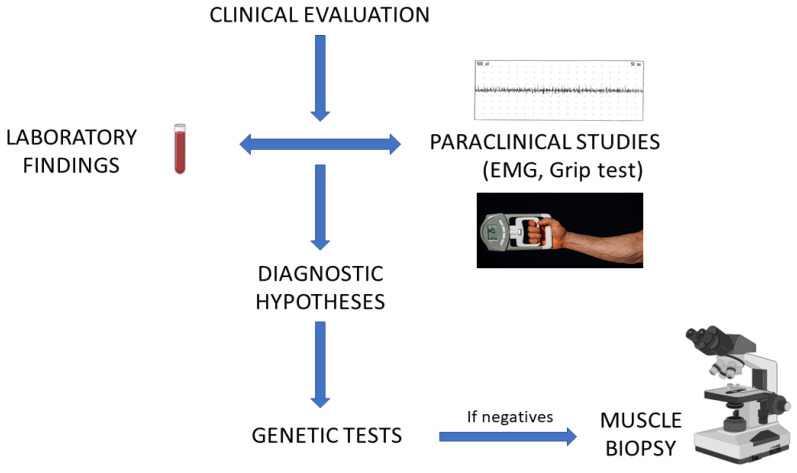
**A semplified diagnostic algorithm for metabolic myopathies.** Created with BioRender.com.

**Table 1 genes-14-00954-t001:** Clinical red flags in metabolic myopathies.

	GLYCOGEN STORAGE DISORDERS								LIPID STORAGE MYOPATHIES					MITOCHONDRIAL MYOPATHIES				
	GSDII	GSDIII	GSDIV	GSDV	GSDVII	GSDXV	PBM1 *	PBM2 *	PCD	CPT II	MADD	NLSD	Lipin-1 Def.	cPEO	Kearn-Sayre	MELAS	MERRF	TK2 Def.
Muscle phenotype																		
Ptosis	x					x								x	x	x		
Ophtalmoplegia	x													x	x	x		x
Macroglossia	x																	
Myalgia	x	x		x	x					x	x		x			x	x	
Proximal muscle weakness	x	x	x				x	x	x		x	x			x	x	x	x
Distal muscle weakness		x						x										
Transient muscle weakness				x														
Rhabdomyolysis	x	x		x	x					x	x		x			x	x	x
Exercise-induced events																		
Second-wind phenomenon				x														
Out-of-wind phenomenon					x													
Exercise intolerance	x	x	x	x	x				x	x	x	x	x	x	x	x	x	x
Extramuscular signs																		
CNS	x		x												x	x	x	x
Liver		x	x				x		x	x								
Heart	x	x	x			x	x		x				x		x	x		x
Respiratory	x																	x
Peripheral neuropathy															x	x	x	

GSD: glycogen storage disorders; PBM: polyglucosan body myopathy; PCD: primary carnitine deficiency; CPT II: carnitine palmitoyl transferase II deficiency; MADD: multiple acyl-CoA dehydrogenase deficiency; NLSD: neutral lipid storage disease; Lipin-1 Def.: lipin 1 deficiency; cPEO: chronic progressive external ophthalmoplegia; MELAS: mitochondrial encephalopathy with lactic acidosis and stroke-like episodes; MERRF: myoclonic epilepsy with ragged red fibers; TK2 Def.: thymidine kinase 2 deficiency. Polyglucosan *.

## Data Availability

Previous version of the paper are accessible on demand. These work did not include experimental data. Figure 1, Figure 2 and Figure 4 are created with BioRender.com. License by Professor Edoardo Malfatti (Premium member).

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
