# Peer review of "Metabolic Myopathies in the Era of Next-Generation Sequencing"

_genes, 2023, doi:10.3390/genes14050954_

Round 1
Reviewer 1 Report
This article provides a very comprehensive overview on metabolic myopathies, highlighting priorities in clinical decision making, giving perspectives on current and potential causative treatment options.
One aspect has, however, been disregarded a little bit, and that is actually what is being promised in the title: role of genetics, especially NGS-based. The reviewer is missing a section on how to interpret variants of unknown significance (e.g. mentioning ACMG criteria), how to prepare for and handle incidental findings, how to select appropriate functional tests for variant pathogenicity, how to conduct proper co-segregation studies. What is specific about NGS-diagnostics in metabolic myopathies? Comparing to other neurogenetic diseases such as CANVAS, CMT1A, or Duchenne muscle dystrophy, are there holes in the whole exome in this field? How would you streamline by phenotype? How much time is normal to pass until finding a diagnosis? Have diagnostic delays been improved and diagnostic gaps been closed in the near past? Are there meaningful gene discoveries in the field, and have registries been established for genotype-phenotype correlations? What about modifier studies?
To not exceed the word count, the first, general part of the manuscript could be comprised a bit more. It would perhaps make sense to depict common metabolic pathways in a figure?
Specific comments:
Abstract: very general, rather like an introduction, should contain more novel content or at least depict conclusions of this work.
English: okay, but not flawless, should be reviewed by a native speaker.
Review:
- on the biopsy part => light microscopic changes in Pompe disease can be quite specific with intracellular glycogen accumulation, making the HE pattern reminiscent of salami sausages
- structure: jumping back and forth a bit, forearm ischemic test comes after biopsies, should be placed alongside clinical examinations or lab tests
- AMPD1(NM_000036.3):c.34C>T(p.Gln12Ter) is more than 20,000 times in GnomAD, >1000 times homozygous. From a genetic point of view, there is no room left for discussion, this is a polymorphism.
- Acid maltase, not maltase acid
- Port wine urine, not porto urine
- “the patient manages and copes with episodes of rhabdomyolysis that will inevitably occur in their life-time” (their instead of its, gender neutral, but not degraded to an object)
- SLC22A5 variants, how can pathogenicity be assessed?
Author Response
In addition to the attachment that contains a revised manuscript, here are the point-by-point responses:
Detailed response to the reviewer of the manuscript “ Metabolic Myopathies in the Era of Next-Generation Sequencing “ by Urtizberea et al.
All the change have been marked in green through the text.
Reviewer #1:
This article provides a very comprehensive overview on metabolic myopathies, highlighting priorities in clinical decision making, giving perspectives on current and potential causative treatment options.
- One aspect has, however, been disregarded a little bit, and that is actually what is being promised in the title: role of genetics, especially NGS-based. The reviewer is missing a section on how to interpret variants of unknown significance (e.g. mentioning ACMG criteria), how to prepare for and handle incidental findings, how to select appropriate functional tests for variant pathogenicity, how to conduct proper co-segregation studies. What is specific about NGS-diagnostics in metabolic myopathies? Comparing to other neurogenetic diseases such as CANVAS, CMT1A, or Duchenne muscle dystrophy, are there holes in the whole exome in this field?
Thanks to the reviewer for this precious advice, we have added a specific section entitled “ NGS in metabolic myopathies: advantages and challenges “ in which we focus on the genetic aspects and also the impact of NGS in the diagnosis of metabolic myopathies, Paragraph 2.3.
- How would you streamline by phenotype?
To simplify the phenotypic classification of the patient with suspected metabolic myopathy, we have added the following scheme in the text. All the elements of the clinical phenotype are graphically associated with the three groups of metabolic myopathies. Font size depends on the frequency of the element in the specific group, the highlighted features are found only in the group they belong to (e.g. distal weakness: glycogen storage disorders).
- How much time is normal to pass until finding a diagnosis?
The time between the onset of symptoms and a diagnosis can be extremely variable; this type of heterogeneity depends on multiple factors among which the most relevant are: the clinical phenotype of the proband (some clinical presentations are highly specific for example “ second wind phenomenon “/ McArdle disease), and the patient assessment to a reference centre with expertise in neuromuscular diseases. We added a sentence in the text: page 4 (lines 363-367)
- Have diagnostic delays been improved and diagnostic gaps been closed in the near past?
Are there meaningful gene discoveries in the field, and have registries been established for genotype-phenotype correlations?
During the last years the increase in scientific knowledge in the field of neuromuscular diseases has certainly reduced diagnostic delay, in particular thanks to two factors: better phenotypic characterization and discovery of new genes associated with metabolic myopathies. To date, there are 30 nuclear genes associated with metabolic myopathies according to the gene table [Benarroch L, Bonne G, Rivier F, Hamroun D. The 2023 version of the gene table of neuromuscular disorders (nuclear genome). Neuromuscul Disord. 2023 Jan;33(1):76-117. doi: 10.1016/j.nmd.2022.12.002. Epub 2022 Dec 6. PMID: 36697115.]
- What about modifier studies?
In the recent years some polymorphisms have been shown to influence the natural history of some metabolic myopathies. For example Ravaglia et al., through an Italian multicentric study showed that polymorphisms in ACE and ACTN3 genes can influence the diaphragmatic dysfunction in LOPD patients. This point has been added to the section “ Advantages and challenges of the use of NGS for metabolic myopathies“.
[ Ravaglia S, Malovini A, Cirio S, Danesino C, De Filippi P, Moggio M, Mongini T, Maggi L, Servidei S, Vianello A, Toscano A, Tonin P, Maioli MA, Parini R, Filosto M, Crescimanno G, Arceri S, Piran M, Carlucci A. Polymorphism in exercise genes and respiratory function in late-onset Pompe disease. J Appl Physiol (1985). 2021 Dec 1;131(6):1762-1771. doi: 10.1152/japplphysiol.00154.2020. Epub 2021 Nov 4. PMID: 34734785.]
- To not exceed the word count, the first, general part of the manuscript could be comprised a bit more. It would perhaps make sense to depict common metabolic pathways in a figure?
Thanks to the reviewer for this suggestion. We have added in the text the following figure, which summarizes the main altered metabolic pathways of skeletal muscle with reference to the respective groups of metabolic myopathies.
- On the biopsy part => light microscopic changes in Pompe disease can be quite specific with intracellular glycogen accumulation, making the HE pattern reminiscent of salami sausages
We added in the text the typical myopathologic light microscopic features of Pompe disease (page 6, lines 464-467)
- Structure: jumping back and forth a bit, forearm ischemic test comes after biopsies, should be placed alongside clinical examinations or lab tests.
We changed ordered the complementary studies as recommended.
- AMPD1(NM_000036.3):c.34C>T(p.Gln12Ter) is more than 20,000 times in GnomAD, >1000 times homozygous. From a genetic point of view, there is no room left for discussion, this is a polymorphism.
We provided this correction.
- Acid maltase, not maltase acid.
We amended the text.
- Port wine urine, not porto urine.
We provided this correction.
- SLC22A5 variants, how can pathogenicity be assessed?
We added in a text a paragraph explaining the validation of SLC22A5 variants associated with PCD. (page 11, lines 733-734)
[Mutlu-Albayrak H, Bene J, Oflaz MB, Tanyalçın T, Çaksen H, Melegh B. Identification of SLC22A5 Gene Mutation in a Family with Carnitine Uptake Defect. Case Rep Genet. 2015;2015:259627. doi: 10.1155/2015/259627. Epub 2015 May 5. PMID: 26075114; PMCID: PMC4436458.]

Reviewer 2 Report
This paper reviews aspects of metabolic myopathies and provides an opinion on testing algorithms.
General comments:
1. There will need to be an extensive re-writing for there are many terminology and semantic errors – just to illustrate a few:
a. Ln 41 – not “uneasy” but difficult or challenging
b. Ln 89 – a prevalence is not elevated – it is higher
c. Ln 98 – these are not “feared” – they are a concern or consideration
d. Ln 170 – very high not “astronomic”
e. Ln 235 – what is RMN???
f. Ln 473 – and
g. Ln – 484 – anamnesis ?? we do not use this term in medical writing
h. Ln -703 – who is her?
2. Pompe disease is technically not a metabolic myopathy – GAA is not used to generate cellular energy and is only used intra-lysosomally to break down glycogen that gets engulfed within the lysosome during cellular autophagy and it is not activated by physical activity to provide energy for the contracting myofibers (Mol Genet Metab. 2012 Nov;107(3):462-8).
3. Ln 68 – 73 – this needs to be re-written – Glycogenolysis is activated within 2 seconds of contraction and the flux through glycolysis is generated by G-6-P that comes from glycogenolysis. In fact, glycogenolysis can continue to provide flux through glycolysis for up to 2 hours and the ratio of fat to CHO is a function of anaerobic threshold, training status, sex and % VO2 max.
4. Ln 109 – this suggests that all patients have some weakness – actually, most do not at all unless it is pseudometabolic or rare late GSD5.
5. Ln 237 – sustained moderate CK elevation needs a full neurological workup by an NMD clinic and not a psychiatrist.
6. The discussion of IOPD and tolerance and all that is not needed in an article on metabolic myopathies – this is outside of the scope and these kids will not present with exercise induced rhabdomyolysis or exercise intolerance but rather with cardiomyopathy and hypotonia. AND Pompe is technically not a disorder that is altering intermediary metabolism – it is the historical lumping into the GSD category that is the issue – it is an LSD and GAA is not part of cellular energy metabolism.
7. Ln 463 – McArdle patients will not be able to sustain 10 min of high-intensity effort – they will have symptoms within a minute or so of high-intensity effort and have to stop of slow down/ease off and then by 10 min and resuming at a lower level then can enter second wind.
8. Ln 700 – are you trying to say that acylcarnitine profiling is not useful in diagnosing CPT2? – that is 100 % incorrect.
9. Overall, the paper is rather scant in terms of reviewing the therapies for each disorder (nutrition and exercise and specific supplements).
10. The authors start the paper by suggesting the use of genome wide or MD panels and yet do not really describe an “approach” to diagnosis.
Minor Points:
Ln 32 – These are not lipidosis but fatty acid oxidation defects – lipidosis is a term for specific LSDs
Ln 33 – what IEM is not due to an enzyme – all enzymes are proteins but not all proteins are enzymes.
Ln 76 – VLCAD and LCAD are not fatty acids – they are enzymes.
Ln – 141 – The presence of multi-organ failure is rare, but can be seen in…
Table 1 – rhabdomyolysis is not spelled correctly and please provide a reference for KSS leading to rhabdomyolysis – if you cannot find on – remove this.
Table 2 – abnormal urine organic acids can be seen in 10-15 % of mitochondrial patients (TCAi, 3-methylgutaconic, etc.).
ENMG – never heard this term – we usually use nerve conduction studies and EMG
Ln 179 – acid phosphatase.
Ln 184 – RRF are VERY RARE to non-existent in FAODs
Ln 204 – not sure what point you are making here – yes sporadic large scale mtDNA deletions are sporadic and not passed on but context?
Ln 227 – we never do parents as a first line – that makes no sense – it is only done to phase variants IF 2 variants are found in an AR disorder OR in children for WES or WGS trio – rarely done in adult metabolic myopathy.
Ln 248 – for mitochondrial do you mean COXPD6? – that is pretty darn rare – most complex I cases are AR.
Ln 399 – MOST LOPD have elevated CK – the normals are usually in late end-stage patients with little to no muscle but at diagnosis it is invariably elevated, albeit mild in some cases.
Ln 620 – ALT and AST when elevated in metabolic myopathies are NOT liver enzymes – the elevation comes from muscle.
Ln 705 – are you saying that a second wind happens in CPT2? It does not.
Ln 739 – weakness of oculomotor muscles = ophthalmoparesis.
Ln 748 – CPEO is a sign defining the disease not a symptom – it is VERY rare for these patients to have rhabdomyolysis.
Ln 771 – sentence structure is wrong – suggest saying that the finding of an elevated lactate can help to rule in disease but a normal lactate dose not rule it out.
Ln 775 – I am not sure that this paper talked about L-carnitine unless levels are low
Author Response
REVIEWER#2:
General comments
- There will need to be an extensive re-writing for there are many terminology and semantic errors – just to illustrate a few :
- Ln 41 – not “uneasy” but difficult or challenging.
Thanks to the reviewer, we provide this correction.
- Ln 89 – a prevalence is not elevated – it is higher.
We made this change.
- Ln 98 – these are not “feared” – they are a concern or consideration.
We changed the term.
- Ln 170 – very high not “astronomic”.
We made this correction.
- Ln 235 – what is RMN???.
We changed the acronyms. Magnetic resonance spectroscopy (MRS)
- Ln 473 – and.
We amended the text.
- Ln – 484 – anamnesis ?? we do not use this term in medical writing.
We used clinical history.
- Ln -703 – who is her?
We corrected the sentence.
- Pompe disease is technically not a metabolic myopathy – GAA is not used to generate cellular energy and is only used intra-lysosomally to break down glycogen that gets engulfed within the lysosome during cellular autophagy and it is not activated by physical activity to provide energy for the contracting myofibers (Mol Genet Metab. 2012 Nov;107(3):462-8).
We agree that Pompe disease is not a metabolic myopathy from a pure biochemical point of view, we added in the text this concept. (page 5, line 428-432)
- Ln 68 – 73 – this needs to be re-written – Glycogenolysis is activated within 2 seconds of contraction and the flux through glycolysis is generated by G-6-P that comes from glycogenolysis. In fact, glycogenolysis can continue to provide flux through glycolysis for up to 2 hours and the ratio of fat to CHO is a function of anaerobic threshold, training status, sex and % VO2 max.
We modified the text accordingly. Paragraph general considerations. Overview of muscle metabolism.
- Ln 109 – this suggests that all patients have some weakness – actually, most do not at all unless it is pseudometabolic or rare late GSD5.
We have reformulated the sentence to clarify this point.
- Ln 237 – sustained moderate CK elevation needs a full neurological workup by an NMD clinic and not a psychiatrist.
This sentence has been removed.
- The discussion of IOPD and tolerance and all that is not needed in an article on metabolic myopathies – this is outside of the scope and these kids will not present with exercise induced rhabdomyolysis or exercise intolerance but rather with cardiomyopathy and hypotonia. AND Pompe is technically not a disorder that is altering intermediary metabolism – it is the historical lumping into the GSD category that is the issue – it is an LSD and GAA is not part of cellular energy metabolism.
We stated that as IOPD manifest as a multisystem disease will not be treated in this review.
- Ln 463 – McArdle patients will not be able to sustain 10 min of high-intensity effort – they will have symptoms within a minute or so of high-intensity effort and have to stop of slow down/ease off and then by 10 min and resuming at a lower level then can enter second wind.
We added a sentence to clearly explain the second wind phenomenon.
- Ln 700 – are you trying to say that acyl carnitine profiling is not useful in diagnosing CPT2? – that is 100 % incorrect.
The sentence has been removed.
9.Overall, the paper is rather scant in terms of reviewing the therapies for each disorder (nutrition and exercise and specific supplements).
We implemented the text with specific nutrition therapeutic approaches and exercise for each group. We also added supplementation approaches for mitochondrial disorders.
- The authors start the paper by suggesting the use of genome wide or MD panels and yet do not really describe an “approach” to diagnosis.
We added the figure 3 in the text in order to simplify the diagnostic algorithm.
Minor Points:
- Ln 32 – These are not lipidosis but fatty acid oxidation defects – lipidosis is a term for specific LSDs.
We provide this correction.
- Ln 33 – what IEM is not due to an enzyme – all enzymes are proteins but not all proteins are enzymes.
We have changed the sentence.
- Ln 76 – VLCAD and LCAD are not fatty acids – they are enzymes.
We corrected the acronyms.
- Ln – 141 – The presence of multi-organ failure is rare, but can be seen in…
We changed this part.
- Table 1 – rhabdomyolysis is not spelled correctly and please provide a reference for KSS leading to rhabdomyolysis – if you cannot find on – remove this.
We removed this sentence and spelled correctly rhabdomyolysis.
- Table 2 – abnormal urine organic acids can be seen in 10-15 % of mitochondrial patients (TCAi, 3-methylgutaconic, etc.). We modified table 2 accordingly.
- ENMG – never heard this term – we usually use nerve conduction studies and EMG.
We changed the text accordingly. acronyms.
- Ln 179 – acid phosphatase.
Corrected.
- Ln 184 – RRF are VERY RARE to non-existent in FAODs.
We modified the text accordingly..
- Ln 204 – not sure what point you are making here – yes sporadic large scale mtDNA deletions are sporadic and not passed on but context?
In this part we want to emphasize the fact that NGS studies from DNA extracted form blood circulating cells cannot reveal the single mtDNA deletions, the mutation is found only in muscle.
- Ln 227 – we never do parents as a first line – that makes no sense – it is only done to phase variants IF 2 variants are found in an AR disorder OR in children for WES or WGS trio – rarely done in adult metabolic myopathy.
We completely agree, we specified that trio analysis is especially made in children when parents are available.
12) Ln 248 – for mitochondrial do you mean COXPD6? – that is pretty darn rare – most complex I cases are AR.
We modified this sentence and we provide the citation of rare cases of AD or X-linked inheritance. [González-Del Angel A, Bisciglia M, Vargas-Cañas S, Fernandez-Valverde F, Kazakova E, Escobar RE, Romero NB, Jardel C, Rucheton B, Stojkovic T, Malfatti E. Novel Phenotypes and Cardiac Involvement Associated With DNA2 Genetic Variants. Front Neurol. 2019 Oct 4;10:1049. doi: 10.3389/fneur.2019.01049. PMID: 31636600; PMCID: PMC6787284.
13) Ln 399 – MOST LOPD have elevated CK – the normals are usually in late end-stage patients with little to no muscle but at diagnosis it is invariably elevated, albeit mild in some cases.
We changed this sentence and better explained the concept.
14) Ln 620 – ALT and AST when elevated in metabolic myopathies are NOT liver enzymes – the elevation comes from muscle.
In this part we are specifically referring to PCD that can presents also with liver disfunction. [Jain S, Kumar K, Malhotra S, Sibal A. Rare case of primary carnitine deficiency presenting as acute liver failure. BMJ Case Rep. 2022 Jul 19;15(7):e247225. doi: 10.1136/bcr-2021-247225. PMID: 35853679; PMCID: PMC9301812.]
15) Ln 705 – are you saying that a second wind happens in CPT2? It does not. In these sentences the second wind is referred to Mc Ardle disease.
We reformulated the sentence.
16) Ln 739 – weakness of oculomotor muscles = ophthalmoparesis.
We provided this correction.
17) Ln 748 – CPEO is a sign defining the disease not a symptom – it is VERY rare for these patients to have rhabdomyolysis.
We provide this correction.
18) Ln 771 – sentence structure is wrong – suggest saying that the finding of an elevated lactate can help to rule in disease but a normal lactate dose not rule it out.
We changed the sentence.
19) Ln 775 – I am not sure that this paper talked about L-carnitine unless levels are low
We mentioned L-carnitine in the part dedicated to the therapeutic approach of mitochondrial myopathies.

Round 2
Reviewer 1 Report
The paper has profited substantially from the revisions and is now ready for publication. I would only recommend to critically double-check for spelling errors or missing commas.
Reviewer 2 Report
The authors have addressed most of my questions. A few tiny suggestions:
1. I would add the following papers regarding exercise and mitochondrial disease:
Taivassalo T, Gardner JL, Taylor RW, et al. Endurance training and detraining in mitochondrial myopathies due to single large-scale mtDNA deletions. Brain 2006;129:3391-3401.
Murphy JL, Blakely EL, Schaefer AM, et al. Resistance training in patients with single, large-scale deletions of mitochondrial DNA. Brain : a journal of neurology 2008;131:2832-2840
2. For nutraceuticals - there have been 3 proper RCTs - best to put in the originals and in your paper suggest "limited number of...":
Rodriguez MC, MacDonald JR, Mahoney DJ, Parise G, Beal MF, Tarnopolsky MA. Beneficial effects of creatine, CoQ10, and lipoic acid in mitochondrial disorders. Muscle Nerve 2007;35:235-242.
Glover EI, Martin J, Maher A, Thornhill RE, Moran GR, Tarnopolsky MA. A randomized trial of coenzyme Q10 in mitochondrial disorders. Muscle & nerve 2010;42:739-748.
Klopstock T, Yu-Wai-Man P, Dimitriadis K, et al. A randomized placebo-controlled trial of idebenone in Leber's hereditary optic neuropathy. Brain : a journal of neurology 2011;134:2677-2686.